# A computational systems approach identifies synergistic specification genes that facilitate lineage conversion to prostate tissue

Flaminia Talos[1,2,3,4,†], Antonina Mitrofanova[2,†], Sarah K. Bergren[1,2,3,4], Andrea Califano[2] & Michael M. Shen[1,2,3,4]

To date, reprogramming strategies for generating cell types of interest have been facilitated by detailed understanding of relevant developmental regulatory factors. However, identification of such regulatory drivers often represents a major challenge, as specific gene combinations may be required for reprogramming. Here we show that a computational systems approach can identify cell type specification genes (master regulators) that act synergistically, and demonstrate its application for reprogramming of fibroblasts to prostate tissue. We use three such master regulators (FOXA1, NKX3.1 and androgen receptor, AR) in a primed conversion strategy starting from mouse fibroblasts, resulting in prostate tissue grafts with appropriate histological and molecular properties that respond to androgen-deprivation. Moreover, generation of reprogrammed prostate does not require traversal of a pluripotent state. Thus, we describe a general strategy by which cell types and tissues can be generated even with limited knowledge of the developmental pathways required for their specification in vivo.

[1] Department of Medicine, Herbert Irving Comprehensive Cancer Center, Columbia University Medical Center, New York, New York 10032, USA. [2] Department of Systems Biology, Herbert Irving Comprehensive Cancer Center, Columbia University Medical Center, New York, New York 10032, USA. [3] Department of Genetics and Development, Herbert Irving Comprehensive Cancer Center, Columbia University Medical Center, New York, New York 10032, USA. [4] Department of Urology, Herbert Irving Comprehensive Cancer Center, Columbia University Medical Center, New York, New York 10032, USA. † Present addresses: Departments of Urology and Pathology, Stony Brook University Medical Center, Stony Brook, New York 11794, USA (F.T.); Department of Health Informatics, Rutgers School of Health Professions, Rutgers Biomedical and Health Sciences, Newark, New Jersey 07107, USA (A.M.). Correspondence and requests for materials should be addressed to A.C. (email: ac2248@cumc.columbia.edu) or to M.M.S. (email: mshen@columbia.edu).

Many studies have demonstrated that expression of lineage-specific regulatory genes can reprogram a mature differentiated cell type into a distinct cell type[1,2]. In many experimental paradigms, transdifferentiation represents a direct conversion from one cell type to another[3–9]. An alternative approach, known as 'primed conversion' or 'indirect lineage conversion', has used transient expression of pluripotency factors to induce a plastic developmental state permissive for respecification of desired cell fates[10–12]. In both reprogramming strategies, however, the specific experimental methods have depended upon prior detailed knowledge of the developmental pathways and regulatory factors that specify the desired cell type[2,13]. Consequently, if only limited information regarding such pathways exist for the cell type of interest, the identification of key regulatory drivers for reprogramming represents a major challenge[14]. Furthermore, this challenge can be particularly difficult when it is likely that combinations of specific regulatory factors are required for reprogramming.

In the case of the prostate, the identification and analysis of drivers of organogenesis is essential for understanding the molecular basis of normal prostate specification, and ultimately for generation of normal human prostate tissue for studies of cancer initiation. However, relatively little is known about the key transcriptional regulators of prostate specification and differentiation in vivo[15]. Therefore, to identify such key regulatory drivers, we have pursued an unbiased computational systems approach that does not depend upon prior literature knowledge.

Our strategy is based on the recent generation of mouse and human prostate interactomes (gene regulatory networks) and the computational methodology employed to identify synergistic drivers of prostate cancer malignancy[16]. In particular, the human prostate interactome is highly relevant for normal prostate biology, as it was generated using a large human patient data set[17] containing a substantial number of expression profiles from normal/benign and low-grade tumour tissues. Furthermore, there is significant cross-species conservation of transcriptional regulatory programs between mouse and human prostate interactomes[16]. Consequently, we anticipated that regulatory genes for normal biological processes would be well-represented in the human prostate interactome.

In our studies, we show that a computational systems approach can identify cell type specification genes that act synergistically, and demonstrate its application for reprogramming of fibroblasts to prostate tissue. Using master regulator analysis[18,19] to interrogate the human prostate interactome, we have identified candidate drivers of prostate specification. We have employed three such master regulators (FOXA1, NKX3.1 and androgen receptor, AR) to generate prostate tissue from mouse fibroblasts, using a primed conversion strategy that involves expression of pluripotency factors, tissue recombination with embryonic urogenital mesenchyme, and renal grafting. Following growth in vivo, the resulting reprogrammed prostate tissue displays appropriate expression of prostate epithelial and stromal markers, can be serially grafted, responds to androgen-deprivation and molecularly resembles control prostate tissue. Interestingly, in contrast with primed conversion of neurons and cardiomyocytes[20,21], the generation of reprogrammed prostate tissue does not appear to require passage through a transient pluripotent state. Thus, our study describes a general strategy by which cell types and tissues can be generated, even with limited knowledge of the developmental pathways required for their specification in vivo.

## Results

### Identification of candidate drivers of prostate organogenesis.
To identify transcriptional drivers (master regulators) of prostate specification and differentiation, we used expression profiles from different stages of prostate organogenesis[22]. Specifically, we generated a prostate 'organogenesis' differential gene expression signature corresponding to the transcriptome changes occurring between the urogenital sinus at embryonic day 16.5, just prior to initial prostatic budding, and the adult prostate at postnatal day 90. To identify candidate transcription factors that drive the phenotypic transition between prostate anlage and mature differentiated prostate tissue, we used the Master Regulator Inference algorithm (MARINa)[18,19] to interrogate the human prostate interactome with the prostate organogenesis signature. Candidate master regulators (MRs) were ranked based on their differential transcriptional activity (DA), which was inferred by the enrichment of their interactome targets in the organogenesis signature (Fig. 1a; Supplementary Data 1).

Since efficient reprogramming is likely to be mediated by the combined activity of multiple regulatory factors, we next used the MARINa algorithm to identify MR pairs that are candidate synergistic regulators of the prostate organogenesis signature. As previously shown[16,19], synergy between two MRs can be computationally inferred by testing whether the enrichment of their shared transcriptional targets in a signature of interest is statistically more significant than enrichment of their individual targets. This analysis identified 11 synergistic pairs, of which 5 pairs contained activated candidate MRs with at least 1 MR differentially expressed in the organogenesis signature (Fig. 1b); we focused on differentially expressed MRs since experimental reprogramming would be most feasible if driven by MR overexpression. Of these five MR pairs, the FOXA1/NKX3.1 pair was the most differentially expressed among all synergistic pairs (Fig. 1c). Notably, these two genes have both been shown to play important roles in prostate organogenesis[23–25], but their potential synergistic interaction during development was not previously investigated. Interestingly, FOXA1 functions as a pioneer transcription factor to recruit steroid hormone receptors to chromatin[26,27], and FOXA1 and NKX3.1 are components of an enhancer complex together with AR on a subset of AR targets[28,29]. Consequently, since AR was also identified as a candidate prostate organogenesis MR, albeit at a lower rank (Supplementary Data 1), we also investigated this factor in our functional studies.

**Reprogramming of fibroblasts to prostate tissue.** Using the candidate master regulators of prostate organogenesis, we employed a primed conversion strategy to generate prostate tissue from mouse embryonic fibroblasts (MEFs), which involves the initial expression of the pluripotency factors OCT4, SOX2, KLF4 and c-MYC (OSKM) and culture conditions that promote epithelial differentiation. Similar primed conversion approaches have been used previously to produce cardiomyocytes, neuronal progenitors and hepatocytes, with OSKM expression used to destabilize the fibroblast differentiated state and lineage-specific factors to specify the desired cell type[10,11,30]. In our strategy, we expressed OSKM factors in MEFs under serum-free culture conditions to generate cells with epithelial properties, followed by lentiviral-mediated expression of FOXA1, NKX3.1 and/or AR and renal grafting in combination with rat embryonic urogenital sinus mesenchyme (UGM) to direct reprogrammed cells towards prostate fate in vivo (Fig. 2a). As a starting population, we used MEFs generated from embryonic day 13.5 dpc (E13.5) limb buds to exclude potential prostate progenitors; in some experiments, we used MEFs from limb buds of $Nkx3.1^{lacZ/+}$ embryos[25] to follow the expression of the endogenous Nkx3.1 gene. For expression of the OSKM pluripotency factors, we used two distinct approaches:

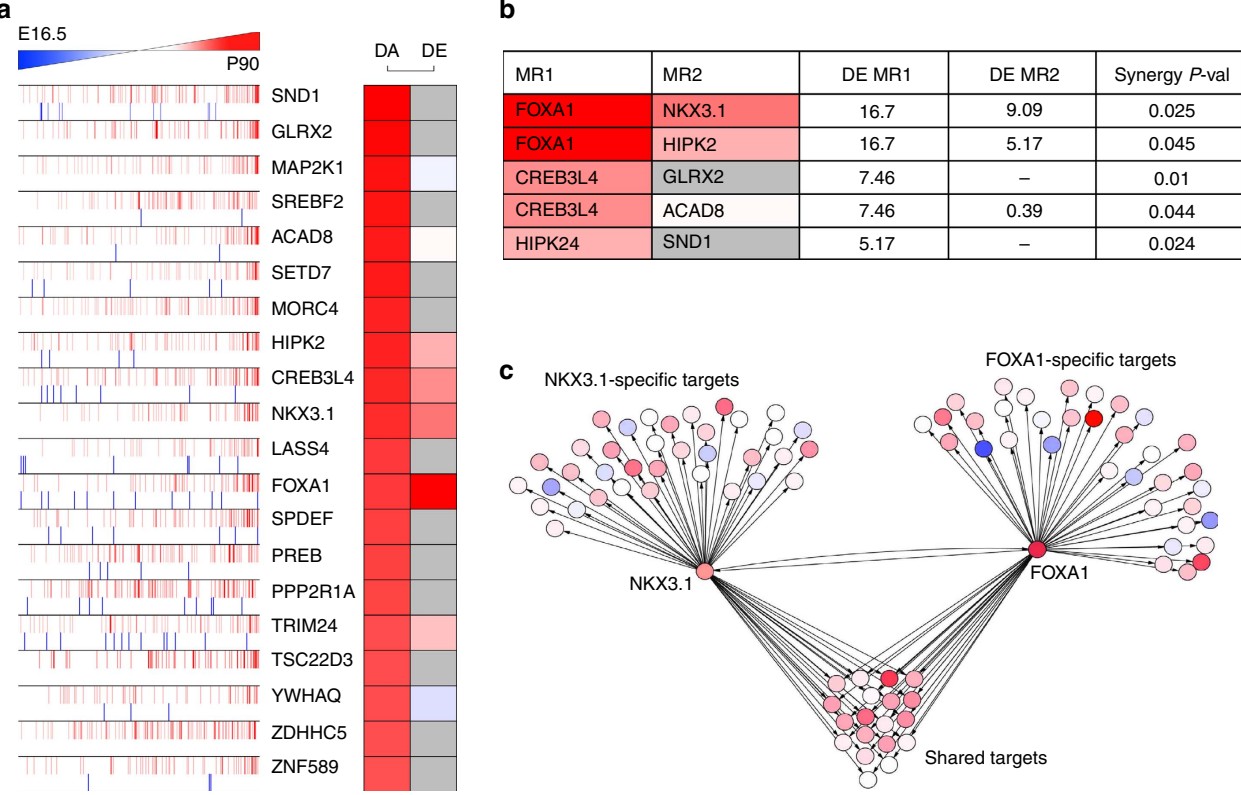

**Figure 1 | Identification of driver genes for prostate specification.** (**a**) List of activated candidate master regulators (MRs) and enrichment profile of their activated (red bars) and repressed (blue bars) target genes on the prostate organogenesis signature, as defined by differential expression analysis of embryonic day 16.5 (E16.5) and postnatal day 90 (P90). Differential activity (DA) is computed as the Normalized Enrichment Score (NES) of the transcriptional targets of a given MR among differentially expressed genes in the signature as computed by MARINa, while differential expression (DE) corresponds to the change in expression of a given MR in the signature. Red indicates increased differential activity/expression, white indicates no significant activity/expression and grey indicates that the gene was not represented on the experimental platform used to generate the signature. (**b**) MR pairs that are predicted to act synergistically regulate the prostate organogenesis signature, including *t*-test and synergy *P* values. (**c**) Network representation of both shared and individual transcriptional targets of NKX3.1 and FOXA1; under-expressed genes and over-expressed genes are represented as blue and red circles, respectively.

(1) retroviral vectors expressing individual OSKM factors constitutively (Supplementary Fig. 1a–d); or (2) mouse dermal fibroblasts (MDFs) established from P0 neonates of *R26R-rtTA; Col1a1-tetO-OSKM* mice, which contain a doxycycline-inducible OSKM transgene[31].

Following retroviral OSKM expression, we observed a rapid morphological alteration in the infected fibroblasts by 48 h, at which time the cells were placed in a serum-free defined basal epithelial medium containing EGF, FGF and dexamethasone in the absence of LIF. Over the course of 16 days, the cultures became enriched for clusters of cells with a cuboidal morphology and expressing epithelial markers, as shown by flow cytometry for EpCAM and CD24 (Fig. 2b), and by immunostaining for epithelial cytokeratins CK5 and CK8, as well as E-cadherin and beta-catenin (Fig. 2c). Thus, we termed these cells 'induced epithelial' cells (iEpt cells). At 6 days after retroviral OSKM expression, we infected the iEpt cells with lentiviruses expressing FOXA1, NKX3.1 and/or AR singly or in combination (Supplementary Fig. 1e–k). Notably, at 10 days following lentiviral-mediated expression of candidate MRs, we found that the iEpt + MR cells showed a pronounced decrease in EpCAM expression, particularly when expressing all three candidate MRs (NKX3.1 + AR + FOXA1 = NAF), whereas control iEpt cells showed increased EpCAM expression by 16 days of culture (Fig. 2b). At the same time, epithelial morphology and

marker expression were retained in the iEpt + NAF cultures (Fig. 2b,c). We observed similar formation of iEpt cells in the case of the doxycycline-induced system, in which MDFs were maintained in the presence of doxycycline to induce pluripotency factor expression and basal epithelial media to stimulate formation of iEpt cells.

Next, we performed tissue recombination of iEpt + MR cells or controls with rat embryonic UGM followed by renal grafting in immunodeficient male nude mice, focusing on iEpt + MR cells generated using OSKM retroviruses. Such tissue recombination approaches have been used extensively to investigate prostate formation as well as assays of prostate reconstitution[32]. In the absence of iEpt or iEpt + MR cells, UGM grafts alone failed to grow, and lacked any epithelial ductal structures (Fig. 2d; Table 1). As a positive control, we performed tissue recombination and renal grafting using dissociated prostate epithelial cells obtained from wild-type adult mice, together with UGM (Fig. 2e,f; Table 1). We found that grafts of iEpt cells lacking MR expression or grafts expressing a single candidate MR could infrequently form small patches of prostate-like tissue (Fig. 2g; Table 1). In contrast, grafts with pairs of candidate MRs could generate prostate-like tissue with high efficiency, although subtle distinctions in histology were apparent in prostate-like tissues obtained from different iEpt + MR combinations (Fig. 2h,i; Table 1). Notably, we found that the

combination of all three candidate MRs in iEpt + NAF grafts efficiently generated prostate tissue that were phenotypically indistinguishable from those found in control grafts (Fig. 2j,k; Table 1). Furthermore, the iEpt + NAF combination was the only one that could form large homogeneous regions of prostate tissue (Fig. 2j).

To characterize prostate differentiation within the tissue grafts, we performed immunostaining with a range of markers using control and iEpt + MR grafts, focusing on the iEpt + NAF grafts, which contained all three candidate MRs. Immunostaining for the luminal markers CK8 and CK18 and the basal markers CK5 and p63 showed that the distribution of basal and luminal

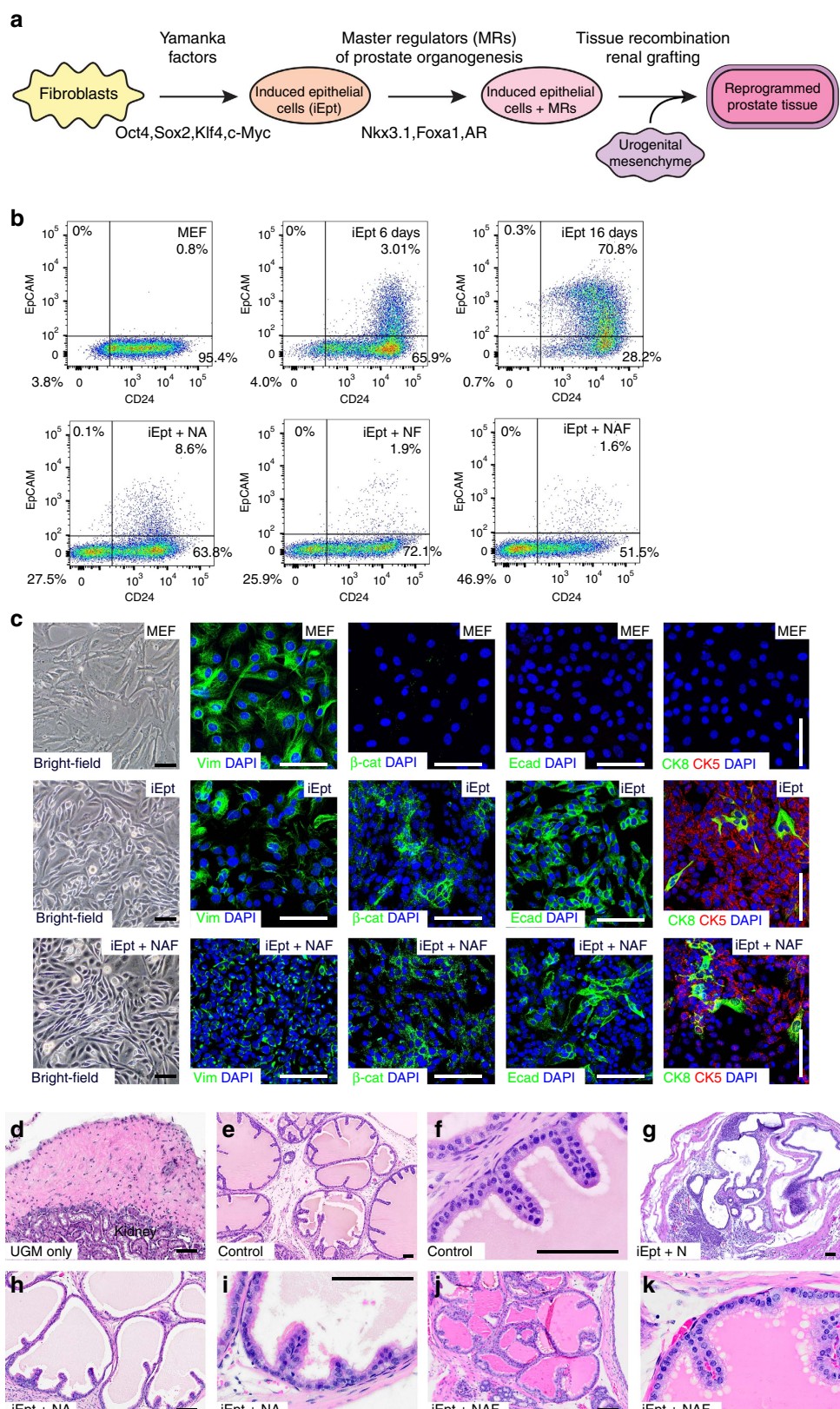

cells was similar between control and iEpt + NAF grafts (n = 6 for each marker; Fig. 3a–d). Immunostaining for synaptophysin also showed that rare neuroendocrine cells could be detected in both control and iEpt + NAF grafts (n = 2; Fig. 3e,f). Importantly, we could readily detect expression of the secretory protein Probasin in both control and iEpt + NAF grafts, indicating the presence of functional differentiated prostate epithelium (n = 9 grafts; Fig. 3g,h).

Moreover, examination of cytokeratin markers and Probasin expression in tissue grafts generated by iEpt cells as well as other iEpt + MR combinations also revealed results consistent with histological analyses. We found that only 1 out of 18 grafts of iEpt cells lacking MR expression contained a small patch of prostate-like tissue, as defined by its ductal structure and expression of Probasin (Fig. 4a–f; Table 1). Grafts of iEpt cells that expressed either Nkx3.1 or Foxa1 as a single candidate MR could only form glandular structures that did not resemble prostate tissue, whereas grafts of iEpt cells expressing AR alone could sometimes generate small patches of prostate-like tissue with limited regions of Probasin immunoreactivity (Fig. 4g–r; Table 1). However, grafts of iEpt cells expressing pairs of candidate MRs (Nkx3.1 and Foxa1, or Nkx3.1 and AR) successfully generated prostate-like tissue with Probasin expression (Fig. 4s–x; Table 1). Similar results were obtained with grafts analysed from doxycycline-inducible iEpt cells expressing Foxa1 and AR (n = 2) (Fig. 4y–a′). Notably, grafts of pluripotent iPSC in the presence (n = 14) or absence (n = 2) of UGM only generated teratomas that lacked prostate tissue (Fig. 4b′–d′). In addition, the MRs were unable to mediate direct conversion of MEFs to prostate tissue in the absence of OSKM factors, as we did not observe epithelial phenotypes in culture or formation of prostate tissue in vivo in renal grafts (Table 1).

Based on our findings that combinations of prostate organogenesis MRs, but not single genes, were successful in the reprogramming assay, we assessed whether the experimentally observed effects of the MRs were synergistic, as had been predicted computationally. For this purpose, we compared the observed combined effects of the MRs to their predicted 'additive' effects using a log-linear model (Supplementary Table 1). This analysis demonstrated a significant synergy of the experimentally observed effects over the predicted additive effects (P values ranging from 0.020 to 0.0023 using a one-sample t-test), indicating that the prostate organogenesis MRs act synergistically in the reprogramming assay.

**Reprogrammed prostate tissue is fully differentiated.** To assess whether the reprogrammed prostate tissue displays the properties of fully differentiated prostate tissue in vivo, we performed additional marker analyses, assessed its response to androgen-deprivation, and investigated its similarity to native prostate at the molecular level. In particular, we could detect strong nuclear expression of AR, Nkx3.1, and Foxa1 in iEpt + NAF

grafts (n = 9 for each marker), consistent with their fully differentiated state (Fig. 3i–n). Moreover, we observed that expression of the stromal markers smooth muscle α-actin (SMA) and vimentin was normal in the iEpt + NAF grafts (n = 3; Fig. 3o,p), indicating that stromal differentiation in the tissue recombinants was properly stimulated by interactions with the prostate epithelium, as is known for wild-type tissue recombinants[33]. Finally, in experiments using MEFs established from $Nkx3.1^{lacZ/+}$ mice, we found that beta-galactosidase staining confirmed the origin of the ductal tissue from the MEFs, as well as activation of the endogenous Nkx3.1 locus (Fig. 3q,r).

After tissue dissociation and recombination with embryonic urogenital mesenchyme, reprogrammed prostate tissue could generate secondary renal grafts containing prostate tissue (Fig. 3s), consistent with the presence of stem/progenitor cells within the primary graft. Using such secondary grafts, we examined whether reprogrammed prostate tissue would respond to androgen-deprivation, which leads to tissue regression associated with massive apoptosis of luminal epithelial cells. Whereas hormonally intact iEpt + NAF secondary grafts (n = 2) displayed strong nuclear AR immunostaining, iEpt + NAF secondary grafts in mice that were castrated and analysed 1 month later (n = 3) showed cytoplasmic AR expression (Fig. 3t,u). Furthermore, cellular proliferation as indicated by Ki67 immunoreactivity was readily detected in the intact iEpt + NAF grafts (10.1 ± 1.9% Ki67-positive epithelial cells, n = 2 grafts), but was abolished in the regressed iEpt + NAF grafts (1.1 ± 0.7% Ki67 positive, n = 3 grafts; P = 0.001 by paired t-test with two-tailed distribution; Fig. 3v,w). In contrast, apoptosis was essentially undetectable by TUNEL staining of intact iEpt + NAF grafts (0.2 ± 0.2% TUNEL-positive epithelial cells, n = 2 grafts), but was readily detected at 3 days after castration (2.2 ± 0.3%, n = 2 grafts; P = 0.01 by paired t-test with two-tailed distribution) and was less evident in the fully regressed iEpt + NAF graft at 1 month after castration (1.1 ± 0.2%, n = 2 grafts; Fig. 3x–z).

To examine the molecular features of the reprogrammed prostate tissue, we performed RNA-seq analyses of MEFs (n = 6), wild-type prostate tissue from 6-week-old C57BL/6 mice (n = 6), iEpt only tissue recombinants (n = 2), and iEpt + NAF recombinants (n = 5), with prostate content of the tissue recombinants confirmed by histology and immunostaining (Fig. 5a–f). Using these RNA-seq data, we determined whether the iEpt + NAF grafts retain expression of the exogenous virally introduced human OSKM as well as organogenesis MRs by comparison of the exogenous human coding regions and the endogenous mouse untranslated regions (UTRs) and coding regions (Supplementary Table 2). These data suggest that maintenance of reprogrammed prostate tissue does not require continuous expression of exogenous MRs.

To assess the molecular similarity of the reprogrammed prostate tissue to normal wild-type mouse prostate tissue,

---

**Figure 2 | Conversion of fibroblasts into prostate tissue.** (**a**) Strategy for production of prostate tissue from fibroblasts by primed conversion. (**b**) Generation of EpCAM + CD24 + cells in MEFs transduced with retroviruses expressing OCT4, SOX2, KLF4, and c-MYC is decreased by infection at 6 d with the indicated combinations of prostate MRs (Nkx3.1 + AR = NA; Nkx3.1 + Foxa1 = NF; Nkx3.1 + AR + Foxa1 = NAF), as analysed by flow-sorting at 16 days. Representative data from four independent experiments are shown. (**c**) Immunostaining for the mesenchymal marker vimentin (Vim) decreases and for the epithelial markers β-catenin (β-cat), E-cadherin (Ecad), cytokeratin 5 (CK5), and cytokeratin 8 (CK8) increases after formation of iEpt cells and infection with MRs (iEpt + NAF); scale bars indicate 50 µm. Representative images from four independent experiments are shown. (**d–k**) Hematoxylin–eosin (H&E) staining of renal grafts obtained from tissue recombination assays with rat embryonic urogenital mesenchyme (UGM); scale bars indicate 100 µm. (**d**) Renal graft performed using UGM alone lacks glandular tissue structures (n = 5 grafts); adjoining kidney tissue is indicated. (**e,f**) Dissociated mouse prostate cells generate prostate tissue in renal grafts (n = 4). (**g**) iEpt cells expressing Nkx3.1 (iEpt + N) generate glandular tissue that does not resemble prostate (n = 3). (**h,i**), iEpt cells expressing Nkx3.1 and AR (iEpt + NA) generate prostate-like tissue (n = 6). (**j,k**), iEpt cells infected with Nkx3.1, AR, and Foxa1 (iEpt + NAF) generate prostate-like tissue that resembles control grafts (n = 18).

| Grafts (+UGM) | Performed | Grown | Prostate tissue | *P* value |
|---|---|---|---|---|
| UGM only | 5 | 0 | 0 | |
| MEFs | 5 | 0 | 0 | |
| MEFs + Nkx3.1 + AR + Foxa1 | 10 | 0 | 0 | |
| iEpt | 22 | 18 | 1 (5.6%) | |
| iEpt + Nkx3.1 | 8 | 3 | 0 | |
| iEpt + Foxa1 | 4 | 4 | 0 | |
| iEpt + AR | 10 | 9 | 3 (33%) | 0.093 |
| iEpt + Nkx3.1 + Foxa1 | 13 | 10 | 4 (40%) | 0.041 |
| iEpt + Nkx3.1 + AR | 21 | 13 | 6 (46%) | 0.026 |
| iEpt + Nkx3.1 + AR + Foxa1 | 56 | 46 | 18 (39%) | 0.013 |

**Table 1 | Summary of renal grafting assays.**

AR, androgen receptor; iEpt, induced epithelial cells; MEFs, mouse embryonic fibroblasts; UGM, urogenital mesenchyme.
Shown is the likelihood of the indicated graft combination to produce any prostate-like tissue compared to iEpt cells alone. *P* value is calculated by two-tailed Fisher's exact test. Note that the iEpt + AR grafts contained only small patches of prostate-like tissue with Probasin immunoreactivity, whereas only the three-factor NAF combination could generate large homogeneous regions of prostate tissue.

we performed Gene Set Enrichment Analysis (GSEA) using a reprogrammed tissue recombinant gene expression signature, defined between profiles of iEpt + NAF grafts versus MEFs, and a normal prostate signature, defined between profiles of wild-type prostate tissue versus MEFs. Comparison of these signatures revealed striking enrichment between the reprogrammed tissue recombinant signature and the normal prostate signature (Fig. 5g). As a further comparison, we performed a second GSEA using a second reprogrammed prostate signature defined between profiles of iEpt + NAF grafts versus iEpt grafts (that did not contain prostate tissue), compared with a normal prostate signature defined between wild-type prostate tissue versus iEpt grafts. Again, we found a strong enrichment between these two signatures (Fig. 5h), indicating the significant molecular similarity of the reprogrammed tissue to normal prostate.

**Reprogramming does not require traversal of pluripotency.** Recent studies have reported that the generation of neuronal progenitors and cardiomyocytes using primed conversion protocols that utilize expression of OSKM require an intermediate 'pluripotent' state marked by transient expression of *Nanog* and *Oct4* (refs 20,21). To investigate whether iEpt and iEpt + MR cells traverse this intermediate state, we used MEFs and MDFs derived from *Oct4-GFP* knock-in mice[34]. Using retroviral expression of OSKM, we could not detect any GFP-positive cells in iEpt culture, even though we could readily detect GFP-positive iPSC colonies grown in ES culture media (Supplementary Fig. 2a,b). We obtained similar results using MEFs derived from a transgenic *Tg(Oct4-GFP)* reporter line[35] (Supplementary Fig. 2c).

As a more stringent test, we employed lineage-tracing to follow the expression of Tomato in MEFs derived from *Tg(Nanog-CreER^{T2}); R26R-Tomato* mice that we have generated (Supplementary Fig. 3a,b). When cultured in the presence of 4-hydroxytamoxifen (4-OHT), activation of *Nanog* transgene expression results in Cre-mediated recombination and expression of the Tomato reporter, which permanently marks *Nanog*-positive cells and all of their progeny, even if they no longer express *Nanog*, thus enabling detection of transient *Nanog* expression. Thus, under standard conditions for generation of iPSC cells, we readily detected Tomato-positive cells that subsequently gave rise to Tomato-positive iPSC colonies (Supplementary Fig. 3c,d). Under iEpt conditions, however, we found that Tomato-positive cells did not represent a significant percentage of the culture at any time point examined, and that their frequency was decreased by the expression of MRs (Fig. 6a,b; Supplementary Fig. 3e). We obtained similar

findings using MEFs from *Nanog-CreER/+; R26R-Tomato/+* mice[20] as well as *Oct4^{CreER/+}; R26R-YFP* mice[36] that were previously utilized for lineage-tracing during neuronal and cardiomyocyte primed conversion (Supplementary Fig. 3f,g).

Notably, following tissue recombination and renal grafting of *Tg(Nanog-CreER^{T2}); R26R-Tomato* iEpt cells cultured in the continuous presence of 4-OHT, Tomato-positive cells were only found as scattered populations (Fig. 6c). After grafting of *Tg(Nanog-CreER^{T2}); R26R-Tomato* iEpt + NAF cells grown with continuous 4-OHT both in culture and *in vivo*, we found that Tomato-positive cells formed ductal structures that did not display prostate features, whereas Probasin-positive prostate ducts were uniformly Tomato-negative (Fig. 6d–f). Thus, these results indicate that primed conversion does not necessarily require transit through an intermediate Nanog-positive state.

## Discussion
Our findings show that Nkx3.1, AR, and Foxa1, which we identified computationally as candidate master regulators of prostate organogenesis are in combination sufficient to confer prostate identity upon fibroblasts. Taken together with previous loss-of-function analyses of these genes[23–25,37,38], we demonstrate that these genes are both necessary and sufficient for prostate specification, and validate their identification as master regulators of prostate organogenesis. Our results are also consistent with the observation that Nkx3.1 and Foxa1 interact with AR to form a complex on a subset of AR transcriptional targets[28,29], although they are likely to have AR-independent functions as well. In addition, recent work has shown that overexpression of Nkx3.1 can respecify mouse seminal vesicle epithelium to prostate[39], indicating that the reprogramming activity of Nkx3.1 is a more general property. Overall, our results confirm the predicted synergy of these factors in prostate reprogramming, providing experimental validation of the computational approaches used in master regulator analysis in a gain-of-function assay.

Our findings also highlight the value of tissue recombination approaches for the generation of reprogrammed tissues containing epithelial and stromal compartments. In particular, the ability of organ-specific mesenchyme to promote expansion of epithelial tissue progenitors has been previously recognized[40]. Furthermore, earlier studies have described the generation of prostate tissue from human embryonic stem (ES) cells in tissue recombination and renal grafting assays with embryonic urogenital mesenchyme or seminal vesicle mesenchyme[41]; however, we were unable to detect formation of prostate tissue

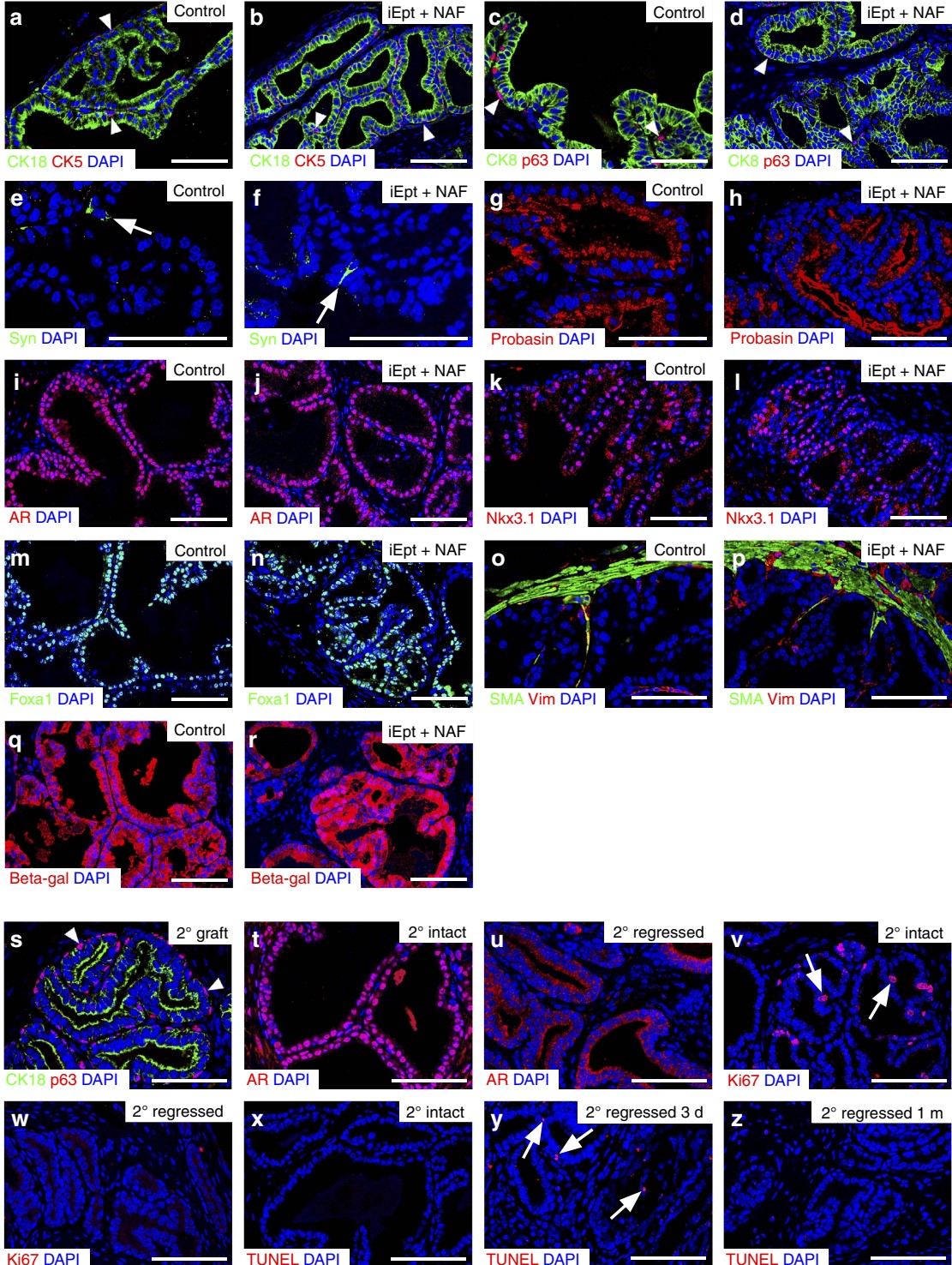

**Figure 3 | Reprogrammed tissue is similar to control prostate tissue.** (a–x) Immunofluorescence analyses of control grafts and reprogrammed iEpt + NAF grafts; Scale bar, 50 µm. (a–d) Luminal cells are positive for CK18 and CK8, whereas basal cells (arrowheads) are positive for CK5 and p63 (n = 6 grafts). (e,f) Detection of rare synaptophysin (Syn)-positive neuroendocrine cells (arrows; n = 2). (g,h) Expression of the prostate-specific secretory protein Probasin (n = 9). (i–n) Expression of AR, Nkx3.1 and Foxa1 (n = 9 for each marker). (o,p) Formation of differentiated stroma expressing smooth muscle actin (SMA) and vimentin (n = 3). (q) Expression of beta-galactosidase in epithelial cells of control prostate graft (n = 1) generated from dissociated prostate epithelial cells from $Nkx3.1^{lacZ/+}$ mice. (r) Expression of beta-galactosidase in prostate epithelium of a graft (n = 6) generated from iEpt + NAF cells derived from $Nkx3.1^{lacZ/+}$ MEFs, demonstrating the donor origin of the epithelial cells as well as expression of the endogenous $Nkx3.1$ locus. (s–z) Secondary grafts of reprogrammed prostate tissue. (s) Expression of the luminal marker CK18 and the basal marker p63 (arrowheads) (n = 3). (t,u) Nuclear localization of AR in the hormonally intact state (n = 2) (t) versus cytoplasmic localization at 1 month after castration (n = 3) (u). (v,w) Immunostaining for the proliferation marker Ki67 in intact (n = 2) (v) and 1 month regressed grafts (n = 3) (w). (x–z) TUNEL staining to visualize apoptotic cells in an intact graft (n = 2) (x) and at 3 days (n = 2) (y) and 1 month after castration (n = 2) (z).

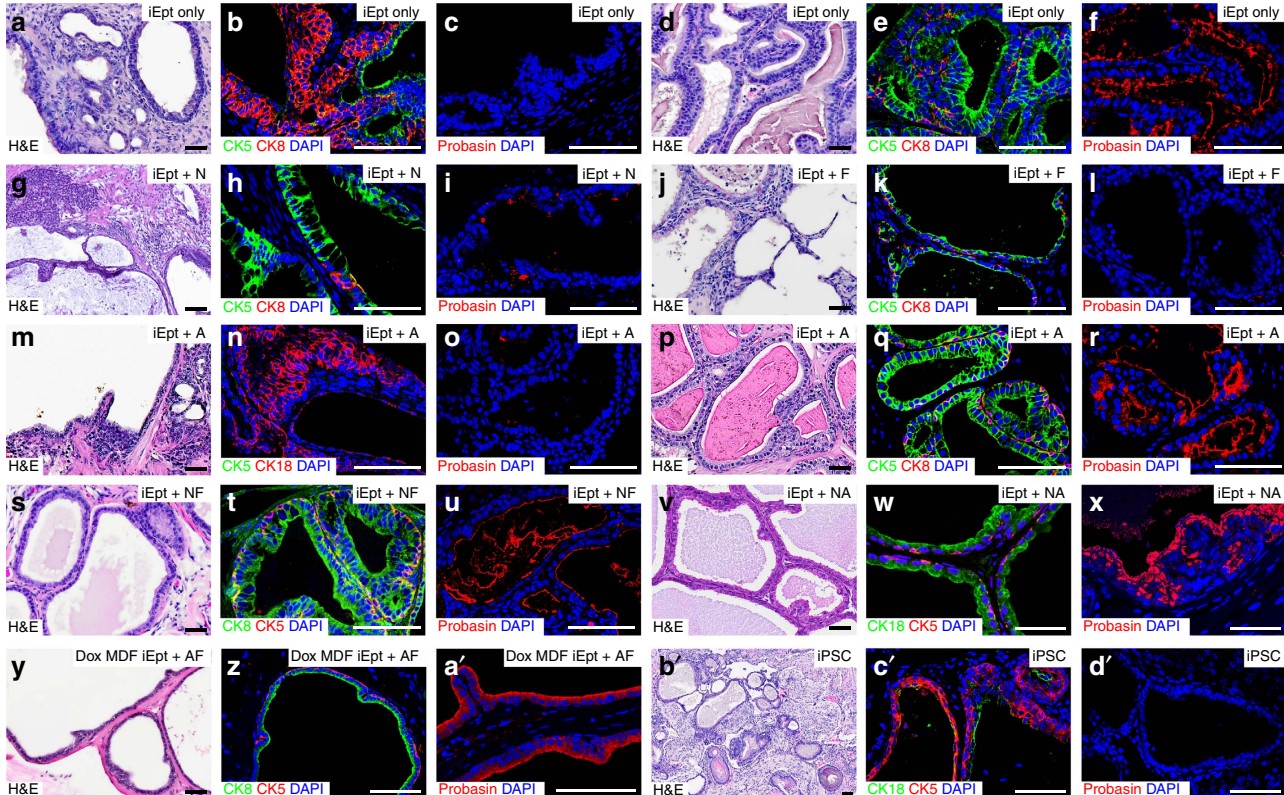

**Figure 4 | Analyses of grafts containing two or fewer candidate master regulators.** (**a–c**) Typical non-prostatic glandular histology in grafts of iEpt cells with rat urogenital mesenchyme (UGM), showing absence of basal-luminal organization ($n = 17$) (**b**) and lack of Probasin-positive secretions ($n = 5$) (**c**). (**d–f**) Small prostate-like region found in 1 of 18 iEpt grafts that is Probasin-positive (**f**). (**g–i**) Non-prostatic histology of graft generated by iEpt cells infected with Nkx3.1 alone (iEpt + N) ($n = 3$). (**j–l**), Non-prostatic histology of graft generated by iEpt cells infected with Foxa1 alone (iEpt + F) ($n = 4$). (**m–r**) Most grafts generated by iEpt cells infected with AR displayed typical non-prostatic histology ($n = 6$) (**m–o**) but 3 out of 9 grafts contained small prostate-like regions with positive Probasin immunoreactivity (**p–r**). (**s–u**) Prostate-like region in graft generated by iEpt cells expressing Nkx3.1 and Foxa1 (iEpt + NF) ($n = 4$). (**v–x**) Prostate-like region in graft generated by iEpt cells expressing Nkx3.1 and AR (iEpt + NA) ($n = 6$). (**y–a'**) Prostate tissue generated from iEpt cells expressing AR and Foxa1 (iEpt + AF) from mouse dermal fibroblasts (MDFs) at passage 1, which were derived from mice engineered with a doxycycline-inducible OSKM cassette ($n = 2$). (**b'–d'**), Teratoma produced by graft of induced pluripotent stem cells (iPSC) together with rat UGM, with glandular regions that lack Probasin expression ($n = 4$) (**d'**). Scale bar, 100 μm (in panels depicting hematoxylin–eosin staining); 50 μm (in panels showing immunofluorescence).

following grafting of mouse iPSC with UGM. More recently, reprogrammed thymic epithelial cells recombined with immature thymocytes and embryonic thymic mesenchyme have been shown to produce functional T cells in renal grafts[42].

Taken together, our studies establish proof-of-concept for a new systems-driven approach for the unbiased and unsupervised identification of specification factors that can drive tissue reprogramming. In contrast with methods that reprogram fibroblasts to a specific cell type in culture, our approach facilitates generation of entire three-dimensional functional tissues *in vivo*. Recent studies have described several distinct computational methods for the prediction of candidate cell type specification genes[43–45]. Notably, given the general requirement for multiple factors to implement reprogramming, the proposed approach for identification of synergistic interactions is especially advantageous, since it reduces the relatively large number of potential specification genes to a handful of computationally prioritized combinations that can be more easily validated experimentally. Indeed, synergy analysis has been highly successful in identifying gene combinations that drive B-cell proliferation, the mesenchymal subtype of glioblastoma, and prostate cancer malignancy[16,18,19]. Our new findings suggest that it will be generally successful for elucidation of reprogramming factor combinations. Thus, we propose that these and related

computational systems approaches will be effective in filling current gaps in our understanding of regulatory factors governing differentiation of desired cell types and tissues.

## Methods

**Computational systems analyses.** We generated the prostate organogenesis signature from published microarray expression data[22]. This signature is represented by a list of genes ranked by their differential expression between E16.5 and P90 ($n = 3$ biological replicates each). The organogenesis signature was used to interrogate a human prostate interactome[16] for master regulator analysis and computational prediction of synergy[18,19]. We performed 1,000 gene permutations to estimate statistical significance ($P$ value) of the Normalized Enrichment Score (NES) for identified MRs. The NES of each MR protein represents the enrichment of its interactome transcriptional targets among genes differentially expressed in the organogenesis signature. Master regulator analysis (MARINa) is available for download at http://califano.c2b2.columbia.edu/software/ as well as through the *viper* package[46] in Bioconductor.

**Mouse strains.** All experiments involving animals were performed according to protocols approved by the Institutional Animal Care and Use Committee at Columbia University Medical Center. *Nkx3.1*^lacZ/+^ mice have been previously described[25], while other lines were obtained from the Jackson Laboratory Induced Mutant Resource, corresponding to *Gt(ROSA)26Sor*^tm1(rtTA*M2)Jae^ *Col1a1*^tm4(tetO-Pou5f1,-Sox2,-Klf4,-Myc)Jae^/J, stock #011011[31], *Oct4-GFP* (B6;129S4-*Pou5f1*^tm2Jae^/J, stock #008214[34], *Tg(Oct4-GFP)* (B6;CBA-*Tg(Pou5f1-EGFP)2Mnn*/J, stock #004654[35], *Oct4-CreER* (B6(SJL)-*Pou5f1*^tm1.1(cre/Esr1*)Yseg^/J, stock #016829[36], *R26R-Tomato* (B6;129S6-*Gt(ROSA)26Sor*^tm14(CAG-tdTomato)Hze^/J,

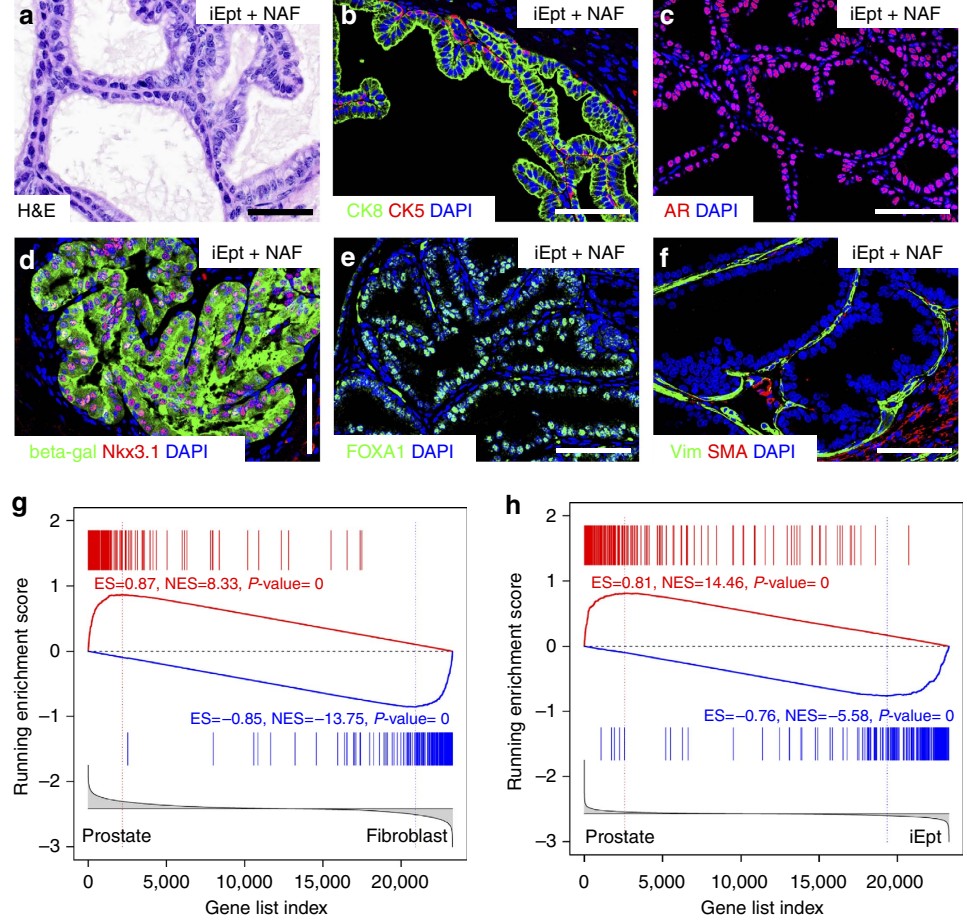

**Figure 5 | Reprogrammed prostate tissue is similar to native prostate at the molecular level.** (**a**–**f**) Representative regions of prostate tissue in iEpt + NAF grafts used for RNA-seq and GSEA analyses; scale bars correspond to 50 μm. (**g**) GSEA comparing signature of reprogrammed prostate tissue defined between iEpt + NAF grafts ($n = 5$) and fibroblasts ($n = 6$) with signature of normal prostate defined between wild-type prostate tissue ($n = 6$) and fibroblasts. (**h**) GSEA comparing signature of reprogrammed prostate tissue defined between iEpt + NAF grafts ($n = 5$) and iEpt grafts ($n = 2$) with signature of normal prostate defined between wild-type prostate tissue ($n = 6$) and iEpt grafts.

stock #007908), and *R26R-YFP* (B6.129X1-*Gt(ROSA)26Sor*tm1(EYFPCos)/J, stock #006148)[47].

To generate *Tg(Nanog-CreER^T2)* mice, BAC recombineering[48] was used to insert a *CreER^T2* cassette precisely at the translation initiation site of the *Nanog* locus on the bacterial artificial chromosome (BAC) vector RP23-406B15 (BACPAC Resources), which has been previously shown to contain essential *Nanog* regulatory sequences[49]. The resulting engineered BAC was utilized for pronuclear microinjection to generate transgenic mice, using standard methods.

**Cell culture.** MEFs were derived from the limbs of E13.5 embryos and dermal fibroblasts (MDFs) were derived from P0-P1 mice using standard protocols. Male and female littermates were pooled to generate fibroblast cultures. We sorted these MEF and MDF cultures against CD45/Ter119/CD31/Mac-1(CD11b)/EpCAM to exclude potential contamination with hematopoietic, endothelial, macrophages and epithelial cells.

MEFs and MDFs were cultured in DMEM supplemented with 10% FBS, $1 \times$ antibiotic/antimycotic (Invitrogen) on 0.1% gelatin-coated tissue culture plates (Millipore). iEpt cells were generated in defined basal epithelial media Cnt-Prime (CellnTec) supplemented with 100 nM dexamethasone (Sigma). iPSCs were maintained in DMEM/F12 medium supplemented with 10% knock-out serum replacement, 1x nonessential amino acids, $1 \times$ GlutaMax, $1 \times$ antibiotic/ antimycotic, 55 μM β-mercaptoethanol (all from Invitrogen), and 1000 U ml$^{-1}$ mouse Leukemia Inhibitory Factor (LIF; Millipore). iPSC colonies were passaged on mitomycin-treated passage 1 wild-type MEFs and maintained in the same media with 10% FBS. Mitomycin was used at a concentration of 10 μg ml$^{-1}$ for 3 h.

**Reprogramming assays.** For studies using OSKM expression from the Rebna retroviral vector[50], viral constructs[51] encoding human *OCT4, SOX2, KLF4* and

*c-MYC* were transfected into Phoenix E cells using Lipofectamine 2000 (Invitrogen), either as a pool or individually in equal concentrations, in which case viral supernatants were mixed just prior to use. After 24 h, transfected Phoenix E cells were treated with 2 μg ml$^{-1}$ puromycin for 2 days and then switched to MEF media. After 12 h in MEF media, viral supernatants were collected every 12 h until cultures became overgrown, followed by splitting 1:4 and another round of puromycin selection. Lin$^{-}$CD11b$^{-}$EpCam$^{-}$ MEFs or MDFs were plated at 250,000 cells per plate in 6 cm dishes and infected with 45-μm-filtered viral supernatants $4 \times$ every 12 h. Cells were switched to basal epithelial medium at 48 h after infection. At 6 days post-OSKM infection, cells were infected $4 \times$ over 2 days with lentiviruses expressing mouse *Nkx3.1-IRES-GFP*, human *AR* and mouse *Foxa1* individually or in combination; lentiviral supernatants were generated as above. The resulting iEpt + MR cells were cultured for an additional 8 days in basal epithelial medium to allow flow-sorting for Nkx3.1-GFP$^{+}$ cells, puromycin selection for Foxa1$^{+}$ cells and blasticidin selection for AR$^{+}$ cells. OSKM-expressing cells that were not infected with lentiviral MRs were used as controls.

For studies using doxycycline-inducible OSKM, passage 0 MDFs obtained from P0 *Col1a1-tetO-OKSM;R26r-rtTA*M2* mice were treated with 2 μg ml$^{-1}$ doxycycline in MEF media for 9 days followed by 8 days of doxycycline withdrawal in basal epithelial media to generate iEpt cells. To generate iEpt + MR cells, lentiviral Nkx3.1, AR and Foxa1 pools (singly or in combination) were used to transduce the cells during days 6 and 7 of doxycycline-induction.

To generate iPSC cells as controls, we used Rebna OSKM retroviruses to infect MEFs derived from E13.5 *Oct4-GFP* embryos and from E13.5 *Tg(Nanog-CreER^T2); R26R-Tomato* embryos, followed by culture in iPSC media in the presence of LIF. Stable undifferentiated GFP-positive or Tomato-positive iPSC colonies were picked at 14–21 days after infection and expanded on feeder cells (mitomycin-treated MEFs) in the presence of LIF. To generate iPSC cells using the doxycycline-inducible system, we treated limb MEFs derived from

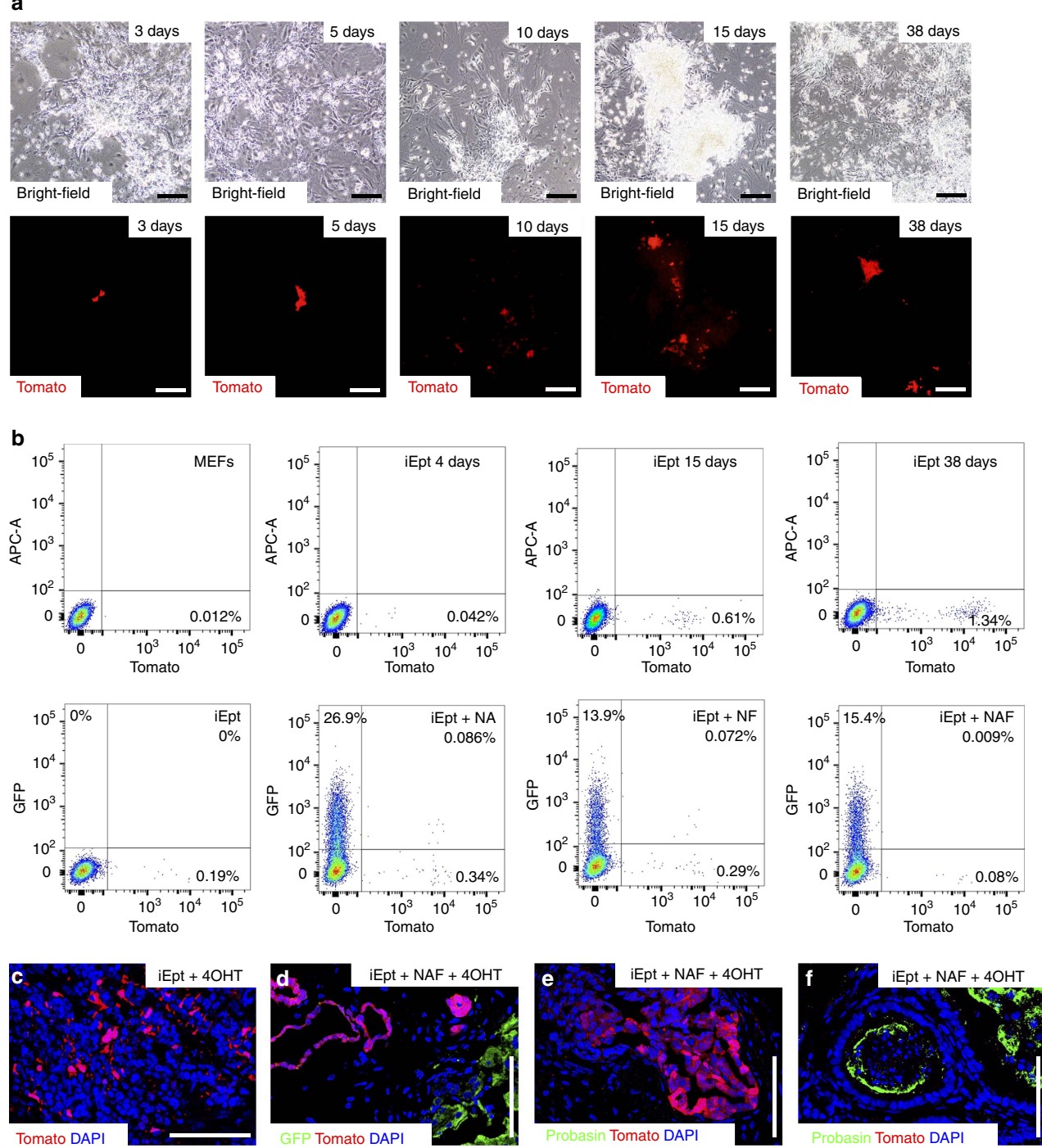

**Figure 6 | Conversion to prostate does not require an intermediate pluripotent state.** (**a**) Direct visualization of Tomato-positive cells obtained by retroviral expression of OSKM in *Tg(Nanog-CreER^{T2}); R26R-Tomato* MEFs, followed by culture in basal epithelial media and 4-OHT (1 μM) for the indicated times; Scale bar, 100 μm. Representative images are shown from three independent experiments. (**b**) Detection of Tomato-positive cells by flow-sorting of *Tg(Nanog-CreER^{T2}); R26R-Tomato* MEFs, iEpt cells cultured in the continual presence of 4-OHT, or iEpt cells infected with lentiviruses for the indicated MRs at 6 days post-infection with OSKM and analysed at day 13 of continuous culture with 4-OHT. Detection of GFP is due to expression of the *Nkx3.1-IRES-GFP* construct used for MR expression. Representative data are shown from 3 independent experiments. (**c**) Direct visualization of Tomato-positive cells in renal grafts (*n* = 3) generated from *Tg(Nanog-CreER^{T2}); R26R-Tomato* iEpt cells cultured continuously in the presence of 4-OHT. (**d**–**f**) Tomato-positive cells in grafts (*n* = 3) generated from *Tg(Nanog-CreER^{T2}); R26R-Tomato* iEpt + NAF cells; 4-OHT was continuously administered to cells in culture as well as to host graft recipients. Scale bar, 50 μm. (**d**) GFP-positive cells (expressing Nkx3.1) contribute to prostate-like tissue which is distinct from regions containing Tomato-positive cells; Scale bar, 50 μm. (**e**) Tomato-positive regions do not form recognizable prostate-like ductal structures. (**f**) Probasin-positive ducts are Tomato-negative.

*Col1a1-tetO-OKSM;R26r-rtTA\*M2* mice for 11 days with 2 µg ml$^{-1}$ doxycycline in iPSC media in the presence of LIF, followed by doxycycline withdrawal for 5 days. Doxycycline-independent iPSC colonies were picked and passaged onto feeder cells for several passages and used in tissue recombination assays.

**Tissue recombination and renal grafting assays.** To perform tissue recombination, 25,000–50,000 iEpt or iEpt + MR cells at 16 days post-infection with OSKM or 1,000–1,500 iPSC cells were mixed with 250,000 dissociated rat urogenital mesenchyme (UGM) cells from E18.5 Sprague-Dawley rat embryos (Charles River), and resuspended in 10 µl of 9:1 collagen/setting buffer solution (10 × Earle's balanced salt solution (Life Technologies), 0.2 M NaHCO$_3$, 50 mM NaOH). Recombinants were cultured overnight in DMEM with 10% FBS and 100 nM DHT, followed by grafting under the kidney capsules of male nude mice (Taconic CrTac:NCr-*Foxn1$^{nu/nu}$*). For each tissue recombination experiment, rat UGM alone was grafted as a negative control to exclude contamination with rat urogenital epithelium. To augment androgen levels, 12.5 mg per 90 day release testosterone pellets (Innovative Research of America) were placed subcutaneously into graft hosts. Renal grafts were harvested for analysis at 6 weeks after implantation.

For control tissue recombinants, mouse prostates (all lobes combined) from WT C57BL/6 and *Nkx3.1$^{lacZ/+}$* mice at 6 weeks of age were collected and dissociated following published protocols[52], followed by tissue recombination as above. A similar procedure was employed to generate secondary grafts by dissociation of primary iEpt + MR grafts followed by tissue recombination with rat UGM and renal graft implantation. Finally, to evaluate the effect of androgen deprivation on the prostate tissue obtained in renal grafts, nude mouse hosts containing iEpt + MRs secondary grafts were castrated and the testosterone pellet removed.

**Analysis of grafts by histology and immunostaining.** Cells cultured in chamber slides were fixed with 4% paraformaldehyde for 30 min, followed by PBS washes and permeabilization-blocking in 0.05% Tween 20/1 × PBS/5% goat serum. For paraffin sectioning, tissues were fixed in 10% formalin for 12–24 h, followed by tissue processing and embedding. Hematoxylin–eosin staining was performed using standard procedures. 5 µm sections were subjected to antigen retrieval by boiling in citrate acid-based antigen unmasking solution (Vector Labs) for 20 min. Sections or fixed cells were incubated with primary antibodies (see antibody suppliers and dilutions in Supplementary Table 3) at 4 °C overnight in humidified chambers. Alexa Fluors (Life Technologies) were used for secondary antibodies. Fluorescence images were acquired using a Leica TCS5 spectral confocal microscope.

Tissue recombinants were systematically sectioned through the entire graft, with four to six 5 µm sections collected every 40–150 µm depending on the graft size, and used for hematoxylin–eosin staining. When prostate-like glandular morphology was observed by bright-field microscopy, we collected 30 serial 5-µm sections for hematoxylin–eosin and immunostaining. Grafts that were entirely composed of prostate tissue were sectioned throughout, with four 5-µm sections collected every 30 µm.

We defined prostate tissue histologically as stratified ductal epithelium forming irregular lumenal projections together with eosinophilic secretions and surrounded by stroma. Tissue architecture was evaluated by immunostaining with basal (CK5) and luminal (CK8) epithelial markers, and confirmed as functional prostate tissue by immunostaining with Probasin, a prostate-specific secretory protein. Immunostaining for beta-galactosidase was used to demonstrate activation of endogenous prostate-specific *Nkx3.1* expression in grafts derived from *Nkx3.1$^{lacZ/+}$* iEpt + MR cells.

**Analysis of experimental synergy.** To determine whether the observed effects of the master regulators (MRs) were synergistic, we compared their experimentally observed combined effects in the reprogramming assay to their predicted 'additive' effects. The predicted additive effects for Nkx3.1, Foxa1, and AR were estimated using a log-linear model based upon observed efficiencies of prostate tissue formation in the reprogramming assay (expressed as %, ranging from 0 to 100%). Log-transformed efficiencies were utilized to fit a linear least-squares curve between control (iEpt) and each test (for example, iEpt + MR) group. Slopes for either (i) a combination of single MRs, or (ii) a pair of MRs combined with an effect of a single MR, were then added to estimate their potential additive effect over the base. The predicted additive effect was then compared to the experimentally observed effect using a one-sample (one-tailed) *t*-test.

**Molecular analysis of gene expression profiles.** RNA-sequencing analyses (30 million single-end reads) for MEFs (*n* = 6), iEpt + NAF grafts (*n* = 5), iEpt only grafts (*n* = 2), and wild-type prostate tissue (*n* = 6) were performed by the Columbia Genome Center using an Illumina Hi-Seq instrument. Computational analyses were done in R-studio 0.99.902, R v3.3.0. Raw counts were normalized using the DESeq package[53]. Differential gene expression signatures were defined using *t*-test statistics between graft and fibroblast or iEpt samples to define the reprogrammed tissue signature, and between prostate and fibroblast or iEpt samples to define the normal prostate signature. Signatures were compared using GSEA[54], where the statistical significance (*P* value) of the enrichment was estimated using 1000 gene permutations.

To determine the expression of exogenous MRs and pluripotency factors in grafts, we performed alignment of the untranslated regions (UTRs) and coding regions of the respective genes to the RNA-seq BAM files of our samples using SAMtools. The UTRs and the coding regions were input according to the UCSC Genome Browser using the human GRCh37/hg19 and the mouse NCBI Build 37/mm9 genome data as references. The presence of exogenous and endogenous transcripts was confirmed by visualization with the Integrative Genomics Viewer (IGV).

**Data availability.** The RNA-seq data generated in this study are deposited in the Gene Expression Omnibus database under the accession code GSE83298. The authors declare that all the data supporting the findings of this study are available within this article and its Supplementary Information, or are available from the corresponding author upon reasonable request.

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

## Acknowledgements

We thank P.-C. Chou, Y. Hu, M. Bhaumik and A. Galli for assistance in the generation and characterization of *Tg(Nanog-CreER$^{T2}$)* mice, C. Abate-Shen, J. Hanna and A. Nemajerova for providing useful reagents, and C. Abate-Shen, C. W. Chua, B. Li, M. Shibata, R. Toivanen and H. Wichterle for advice and comments on the manuscript. This work was supported by grants from the NIH (A.C., M.M.S.), by a Prostate Cancer Foundation Young Investigator award (A.M.), and a Urology Care Foundation Research Scholar Award (F.T.).

## Author contributions

F.T. carried out experiments, A.M. performed bioinformatic analyses and S.K.B. performed renal grafting. F.T. and M.M.S. designed the overall study, A.C. and M.M.S. supervised the data analysis, and F.T. and M.M.S. wrote the manuscript. All authors provided discussion and comments on the manuscript.

## Additional information

**Competing financial interests:** The authors declare no competing financial interests.

