## [Peer Review File · Nature Communications]

Reviewer #1 (Remarks to the Author)

I think the authors have reasonably revised the manuscript. The paper showed a successful example of computer based reprogramming strategy based on sufficient biological data. Their results are informative for broad readers of Nature Communications.

Reviewer #2 (Remarks to the Author)

Overall, the manuscript is well done and convincing. There is also sufficient novelty to the general and prostate communities.

There are several points that needs clarification and should be addressed. Given the additional space allotment in Nature Communications, it would be nice to include more detailed characterizations and move some data into the main figures. I find extended Figs 2 & 3 very important and it would be better if they were moved to the main figures.

1. As noted, the authors used constitutive lentiviruses to express both the Yamanaka factors as well as NAF. The expression of the exogenous and endogenous factors of the grafts were not systemically explored, with exception of the Nkx3-1-LacZ knockin. This is in fact easy to do as RNA-seq of the grafts can easily distinguish the transgene which do not have 5' and 3' UTRs. This analysis would be very useful for all 7 factors introduced.

2. For figure 3y and 3z, the analysis is an expected results and offers little value. The authors compare mouse prostate, NAF grafts, both in vivo, and MEFs in culture. THEY show that compared to MEFs, the NAF grafts are more similar to prostate. This is an obvious result and offer no insights to the role of NAF. The differences include in vitro vs in vivo, epithelial vs stromal. Authors show that iEpts with just Yamanaka factors cause MET to MEFs.

To determine the role of NAF, it would be necessary to compare iEpt grafts (Extended Fig 2a) which show epithelial structures which is not prostate like with iEpt + NAF grafts.

Short of that, an analysis of the differences between NAF grafts and mouse prostates should be performed.

3. technically, what is the efficiency of introduction of Yamanaka factors? Are they 4 independent lentiviruses and how many cells have all 4 infected vs. less than that? Paper notes puromycin for Yamanaka factors but also notes puromycin for AR.

Reviewer #3 (Remarks to the Author)

This is an excellent and interesting piece of work, however this review primarily concerns the computational analysis of TFs, and does not attempt to make a recommendation on the manuscript as a whole.

Major comments

The title of the manuscript contains the word 'synergistic', meaning that the whole is greater than the sum of the parts. Both previous reviewers had between them several questions around this issue. From what I can tell, in three papers of the authors' own work which are cited, the same computational approach was used to identify potentially synergistic TFs. In the previous papers this was followed up by an experimental program to back up in some way the evidence for synergy of a specific example. In this paper however the mechanism is not uncovered and the evidence for synergy (versus simple complementarity -the sum of the parts-) is weak. The authors argue in this paper that the previous work shows that synergy can be inferred computationally, which although true, does not constitute much evidence; the computational approach will generate targets that

are possibly synergistic but not necessarily so.

The computational prediction identifies TFs which overlap in their target genes and have enriched differential expression in a certain way. The effect of these two TFs together could be synergistic or simply complementary. To suggest that pairs of candidates found in this way are identified as synergistic is going beyond what the cited papers have established.

If it were not in the title of the manuscript and central to the way the otherwise excellent work is presented, this may not be such a major comment.

Minor comments

The authors may choose to either remove or substantiate the claim of synergy. However if the claim of synergy remains as central to the presentation, then the scientific argument needs to be made explicitly in the paper and sufficient detail provided in the methods. Both of the previous referees struggled with this, as would a reader, and I had to expend significant effort understanding the three previous publications.

Reviewer #1:

I think the authors have reasonably revised the manuscript. The paper showed a successful example of computer based reprogramming strategy based on sufficient biological data. Their results are informative for broad readers of Nature Communications.

Response: We thank this reviewer for his/her interest and support.

Reviewer #2:

Overall, the manuscript is well done and convincing. There is also sufficient novelty to the general and prostate communities.

There are several points that needs clarification and should be addressed. Given the additional space allotment in Nature Communications, it would be nice to include more detailed characterizations and move some data into the main figures. I find extended Figs 2 & 3 very important and it would be better if they were moved to the main figures.

Response: We thank this reviewer for this suggestion. In this resubmission, we have extensively revised the figure presentation, and have moved the contents of the previous Extended Figures 2 and 3 into the main figures, as noted above.

- 1) *As noted, the authors used constitutive lentiviruses to express both the Yamanaka factors as well as NAF. The expression of the exogenous and endogenous factors of the grafts were not systemically explored, with exception of the Nkx3-1-LacZ knockin. This is in fact easy to do as RNA-seq of the grafts can easily distinguish the transgene which do not have 5' and 3' UTRs. This analysis would be very useful for all 7 factors introduced.*

Response: We thank this reviewer for this suggestion. In our revised manuscript, we have used the RNA-seq data to investigate the expression of exogenous and endogenous master regulators as well as the OSKM pluripotency factors. These data show that 3 out of 5 reprogrammed prostate tissue samples (iEpt+NAF) do not express exogenous NAF, consistent with the analysis of endogenous *Nkx3.1-lacZ* expression in Fig. 3r. The expression of exogenous OSKM is more variable in these samples, but again is consistent with a lack of requirement for their expression in graft maintenance, consistent with the analysis of grafts using doxycycline-inducible OSKM in Fig. 4y-a'. These data are described on pp. 10-11 and are included in Supplementary Table 3.

- 2) *For figure 3y and 3z, the analysis is an expected results and offers little value. The authors compare mouse prostate, NAF grafts, both in vivo, and MEFs in culture. They show that compared to MEFs, the NAF grafts are more similar to prostate. This is an obvious result and offer no insights to the role of NAF. The differences include in vitro vs in vivo, epithelial vs stromal. Authors show that iEpts with just Yamanaka factors cause MET to MEFs.*

To determine the role of NAF, it would be necessary to compare iEpt grafts (Extended Fig 2a) which show epithelial structures which is not prostate like with iEpt + NAF grafts.

Short of that, an analysis of the differences between NAF grafts and mouse prostates should be performed.

Response: To address this important critique, we have performed additional RNA-seq analyses of iEpt grafts that did not contain prostate tissue. Using these new RNA-seq data, we have performed gene set enrichment analysis (GSEA) to compare a reprogrammed prostate gene expression signature, defined by expression profiles of iEpt+NAF grafts versus iEpt grafts, with a native prostate expression signature, defined by expression profiles of wild-type prostate tissue versus iEpt grafts. The comparison of these signatures revealed a strong enrichment between the reprogrammed prostate signature and the native prostate signature, indicating the significant molecular similarity of the iEpt+NAF grafts to normal prostate. These data are described on p. 11 and shown in Fig. 5h.

- 3) *technically, what is the efficiency of introduction of Yamanaka factors? Are they 4 independent lentiviruses and how many cells have all 4 infected vs. less than that? Paper notes puromycin for Yamanaka factors but also notes puromycin for AR.*

Response: As described in the Methods section of our manuscript, we used Rebna retroviral vectors for expression of OSKM (Yamanaka factors) in fibroblasts. The Rebna retroviral vectors contain a puromycin selection cassette that is only expressed in the transfected Phoenix E cells, and the resulting retroviruses do not carry this selection marker after infection of fibroblasts. Thus, we used puromycin to select for Phoenix E cells transfected with OSKM prior to collecting viral supernatants, ensuring production of high-titer viral stocks and consequent high infection efficiency of the fibroblasts. Since the resulting Rebna retroviruses do not contain a puromycin selection cassette, we were able to use puromycin again to select for lentiviral infection of prostate organogenesis master regulators such as AR.

To assess the efficiency of retroviral expression of OSKM, we examined OSKM expression at 2-4 days post-infection in multiple independent experiments (n=4). We determined the infection efficiencies for each individual factor as 85.2% \pm 5.9% for OCT4, 84.6% \pm 6.6% for SOX2, 84.4% \pm 2.9% for KLF4, and 90.8% \pm 6.4% for cMYC. Based on co-immunostaining experiments, we also determined that 82.4% \pm 1.7% of the cells were co-infected with OCT4 and KLF4. This description and representative images of these infections have been included in Supplementary Figure 1a-d.

Due to the incompatibility of several of the antibodies used for co-immunofluorescence experiments (as they were raised in the same species), we are unable to perform a direct assessment of the co-expression of all four OSKM factors. However, based upon our observed efficiencies for expression of the single factors and for OCT4 and KLF4 co-expression, we can calculate an estimated minimal percentage of cells co-expressing all four factors as 43.1%, and an estimated maximal percentage of cells co-expressing all four factors as 84.1%. This analysis is described in the legend to Supplementary Figure 1.

Reviewer #3:

This is an excellent and interesting piece of work, however this review primarily concerns the computational analysis of TFs, and does not attempt to make a recommendation on the manuscript as a whole.

Major comments

The title of the manuscript contains the word 'synergistic', meaning that the whole is greater than the sum of the parts. Both previous reviewers had between them several questions around this issue. From what I can tell, in three papers of the authors' own work which are cited, the same computational approach was used to identify potentially synergistic TFs. In the previous papers this was followed up by an experimental program to back up in some way the evidence for synergy of a specific example. In this paper however the mechanism is not uncovered and the evidence for synergy (versus simple complementarity -the sum of the parts-) is weak. The authors argue in this paper that the previous work shows that synergy can be inferred computationally, which although true, does not constitute much evidence; the computational approach will generate targets that are possibly synergistic but not necessarily so.

The computational prediction identifies TFs which overlap in their target genes and have enriched differential expression in a certain way. The effect of these two TFs together could be synergistic or simply complementary. To suggest that pairs of candidates found in this way are identified as synergistic is going beyond what the cited papers have established.

If it were not in the title of the manuscript and central to the way the otherwise excellent work is presented, this may not be such a major comment.

Minor comments

The authors may choose to either remove or substantiate the claim of synergy. However if the claim of synergy remains as central to the presentation, then the scientific argument needs to be made explicitly in the paper and sufficient detail provided in the methods. Both of the previous referees struggled with this, as would a reader, and I had to expend significant effort understanding the three previous publications.

Response: We thank the reviewer for this insightful comment. In response, we have performed a new analysis to demonstrate that our experimental data support the conclusion that the three master regulators of prostate organogenesis (Foxa1, Nkx3.1, and AR) are indeed acting synergistically in the reprogramming assay.

To determine whether the observed effects of the master regulators (MRs) were synergistic, we compared their experimentally observed combined effects in the reprogramming assay to their predicted “additive” effects. The predicted additive effects for Nkx3.1, Foxa1, and AR were estimated using a log-linear model based upon observed efficiencies of prostate tissue formation in the reprogramming assay (expressed as %, ranging from 0% to 100%). Log-transformed efficiencies were utilized to fit a linear least-squares curve between control (iEpt) and each test (*e.g.*, iEpt+MR) group. Slopes for either (*i*) a combination of single MRs, or (*ii*) a pair of MRs combined with an effect of a single MR, were then added to estimate their potential additive effect over the base. The predicted additive effect was then compared to the experimentally observed effect using a one-sample (one-tailed) t-test, which demonstrated a significant synergy of the experimentally observed effect over the predicted additive effects. This analysis is described on p. 9 and pp. 19-20 and the results are shown in Supplementary Table 2.

Reviewer #2 (Remarks to the Author)

The extensively revised manuscript has addressed all of my concerns and I believe it will be a significant contribution to both prostate development and prostate cancer fields.

Reviewer #3 (Remarks to the Author)

The main concern that I raised has been addressed by the authors, and I hope that the manuscript is improved for it. I think however that on page 9 where it is declared significant (and supp table 2 is referred to) that it would be further improved by making it a quantitative statement there, rather than the present qualitative statement. This requires only adding a few words and will save the superficially curious reader a trip to the supplementary data.

In response to the remaining comment by Reviewer #3, we have made the following change.

The main concern that I raised has been addressed by the authors, and I hope that the manuscript is improved for it. I think however that on page 9 where it is declared significant (and supp table 2 is referred to) that it would be further improved by making it a quantitative statement there, rather than the present qualitative statement. This requires only adding a few words and will save the superficially curious reader a trip to the supplementary data.

Response: We thank this reviewer for this suggestion. In the revised text, we have modified the relevant sentence on p. 9 to state that: *“This analysis demonstrated a significant synergy of the experimentally observed effects over the predicted additive effects (p-values ranging from 0.020 to 0.0023 using a one-sample t-test), indicating that the prostate organogenesis MRs act synergistically in the reprogramming assay.”* We believe that this modification addresses the reviewer’s comment.

We hope that this revised manuscript is now suitable for publication in *Nature Communications*. Thank you for your careful consideration.